# Green Bond Issuance and Peer Firms' Green Innovation

**Xia Wu** [1,2], **Danlu Bu** [1], **Jun Lian** [2,*] **and Yanping Bao** [3]

1   School of Accounting, Southwestern University of Finance and Economics, Chengdu 610074, China
2   School of International Business and Management, SiChuan International Studies University, Chongqing 400031, China
3   School of Accounting, Chongqing Finance and Economics College, Chongqing 401320, China
*   Correspondence: lianjun@sisu.edu.cn

**Abstract:** Based on the realistic background of the rapid development of China's green bond market, this paper uses the data of China's non-financial listed companies from 2010 to 2020 to examine the impact of green bond issuance on peer firms' green innovation. The results show that the issuance of corporate green bonds can significantly promote the quantity and quality of peer firms' green innovation, and this promotion effect is sustainable. The heterogeneity test shows that when the issuer of green bonds is an industry leader or the issuer is highly concerned by the media, the green innovation promotion effect of peer firms is more significant. Similarly, when the issuer and the peer firm are close competitors or in the same board network, the peer firm has a higher level of green innovation. It is further found that the green innovation behavior adopted by peer firms can significantly improve their environmental performance. The article indicates that the issuance of corporate green bonds can produce a good spillover effect of green innovation in the industry, which is conducive to China's strategic goal of "carbon neutrality, carbon emission peak".

**Keywords:** green bond; green innovation; peer firm; environmental performance

## 1. Introduction

According to the latest version of the Climate Bond Standard issued by the Climate Bond Initiative (CBI), green bonds refer to a bond where the proceeds will be exclusively applied to finance or re-finance, in part or in full, new or existing eligible green projects, which are aligned with the four core components of the Green Bond Principles (Check the Climate Bond Standard for details of the four core components. https://www.climatebonds.net/files/files/climate-bonds-standard-v3-20191210.pdf (accessed on 10 December 2022)). Similar to traditional fixed-income securities, enterprises can raise capital through green bonds. In addition, green bonds need to report, in detail, the use of the raised funds in green projects and the "green" nature of the projects [1]. Moreover, green bonds aim to produce positive environmental benefits. Previous studies have shown that green bonds play a role in reducing carbon dioxide emissions [2], improving air quality [3], and improving environmental performance [4], although there are some greenwashing behaviors in the green bond market [5].

In 2007, the European Investment Bank (EIB) issued the first "Climate Awareness Bond", which marked the rise of the green bond market [6]. After this, the green bond market developed slowly until 2013, when the market began to grow rapidly and entered an "exciting period" [7]. Although the green bond market has developed rapidly in recent years, due to its late start, the issuance of green bonds is less than 1% of the global cumulative bond issuance [8]. In addition, the statistical results of this research sample show that only 63 non-financial listed companies in China have successfully issued green bonds by the end of 2020, which is less than 1.5% of the total number of listed companies in China. Therefore, it is important to ask, how can such a small amount of green bonds promote overall environmental improvement? A reasonable explanation is that the issuance

of green bonds not only affects the pro-environmental behavior of the issuer [2] but also drives peer firms to take more measures that are beneficial to environmental protection [3], thus resulting in better overall environmental performance and social welfare.

Throughout the existing literature, scholars mainly study green bonds from two aspects: issue pricing and market reaction. Regarding green bond issuance pricing, most studies believe that green bonds can signal enterprises' ecologically sustainable development to the market and enhance the ESG value of issuers. Investors who are concerned about environmental protection may be willing to sacrifice a certain expected income to buy green bonds, resulting in green premium [5,9–13]. In research on the market reaction of green bonds, scholars have found that the stock market has a positive response to green bonds issuers by using the event study method [2,14–16]. One stream of research has also confirmed that green bonds can produce a positive environmental performance [2,4] and financial performance [17]. At the present stage of the development of the green bond market, the corporate green bonds issuers have explored new financing ways for other enterprises; they are the "pioneers" of the green bond market, and their successful practices will inevitably be learned and referenced by peer firms, thus affecting the behavior of peer companies. Wu et al. (2022) [3] found that enterprises issuing green bonds will drive the same industry companies to take more actions that benefit environmental protection. These behaviors are recognized by investors, thus reducing the bond financing costs of other enterprises in the same industry. However, up to now, there is no article to explore the impact of green bonds issued by enterprises on the green innovation behavior of peer companies. This paper will make a supplement in this field.

We use the non-financial listed companies in China from 2010 to 2020 as samples to test the impact of green bonds on the green innovation of peer firms. Since the Green Bond Issuance Guidelines were released in 2015, China's green bond market has developed rapidly. By the end of 2020, China has issued a total of USD 258.8 billion of green bonds in domestic and overseas markets and has become the second largest green bond issuer (The data come from the China Green Bond Market Report 2021. You can access https://www.chinabond.com.cn/cb/cn/yjfx/zzfx/nb/20220704/160692668.shtml (accessed on 10 December 2022) for the original text). The Wind database is the largest, most complete, and most widely used financial database in China. It contains all the announcements of listed companies, stock funds, bond data, financial laws, and regulations in the China stock market. The Wind database comprehensively includes the announcement date, circulation, issue price, basic information of issuers, and other data on green bonds issued by China enterprises, which meets the data requirements of this study. Suppose enterprises in an industry have successfully issued green bonds. In that case, we take the companies that have not issued green bonds in this industry as the treatment group and use the propensity score matching (PSM) method to match the appropriate control groups in other industries. Then, we build a difference-in-difference (DID) model to test the impact of green bonds on the green innovation of peer firms. Whether enterprises issue green bonds is not random, and PSM can alleviate the problem of selectivity deviation in the model. The DID model can effectively evaluate the implementation effect of green bonds, thus alleviating the endogenous problem of the model. From this data set, we find that the issuance of corporate green bonds can significantly improve the quantity and quality of peer firms' green innovation, which is manifested in the significant increase in the total number of green patent applications, green invention patent applications, and green patent citations of peer firms. This result is robust in several different specifications, including, but not limited to, changing the PSM matching method, the explained variables being delayed by one period, using the Tobit model to re-estimate the results, conducting a placebo test, and excluding the influence of industry policies. The results from these specifications are similar to the baseline.

The premise that the DID model is effective is that the enterprises in the treatment group and the control group have a parallel trend before the policy impact, which we have

confirmed. We find that the green bond has a dynamic and sustained role in promoting the green innovation of peer firms.

Next, we are interested in what kind of green bond issuers can produce better green innovation spillover effects. We find that when the green bond issuer is an industry leader or the issuer is highly concerned by the media, the green innovation promotion effect of peer firms is more significant. Similarly, when the issuer and the peer firm are close competitors or in the same board network, the peer firm has a higher level of green innovation.

We further explored whether the green innovation behavior adopted by the peer firms can produce positive environmental performance. We found that the green innovation behavior of peer firms can significantly increase their probability of obtaining environmental recognition or other positive evaluation, improve their environmental responsibility scores, and promote them to pass the ISO14001 certification. Generally speaking, our research shows that peer firms improve their environmental performance by upgrading the quantity and quality of green innovation after enterprises issue green bonds.

This research clarified the mechanism of green bonds on the green innovation behavior of peer companies, explored the specific path of green bonds to produce environmental performance, and answered an important question in practice, that is, how does the green bond market support the sustainable development of economy and ecology in the initial stage of development?

Moreover, this study expands and enriches the related literature on green bonds. The existing research mainly focuses on the issues of the pricing of green bonds [9–11,13,18], stock market reflection [2,16], environmental performance [2,4], financial performance [17], links with other financial products [19,20], and factors affecting green bonds [21,22]. So far, there is no literature that studies the innovation spillover effect of green bonds. We supplement this research system by studying the impact of green bond issuance on the green innovation of peer firms.

Furthermore, this study also contributes to the literature on the influencing factors of green innovation. The driving factors of green innovation have been widely considered in academic circles. The existing research mainly focuses on the following aspects: First, environmental regulation, mainly including environmental tax collection, government subsidies, emission trading, environmental information disclosure, environmental decentralization, environmental interview, environmental policy uncertainty, carbon emission regulation, government environmental expenditure, central environmental protection inspector, and other factors [23–26]. Second, the enterprise's organizational structure, mainly including factors, such as company resources, corporate governance mechanism, and senior management characteristics [27–31]. Third, the external environment, mainly including the digital economy, intellectual property protection, air pollution, Internet development, media attention, and other factors [32,33]. At present, the research on the promotion path of green innovation lacks the realistic consideration of the support of financial instruments. Although a few pieces of literature have studied the promotion of green finance to green innovation from the perspective of green credit, the research on other green financial instruments represented by green bonds is lacking.

In addition, this study also adds to the literature on the role of peers in firms' decision making. Previous literature has studied the influence of executive compensation [34], stock split [35], dividend policy [36], M&A decision [37], investment decision [38,39], etc., on peers' decision making, and we have supplemented the literature on the influence of financing decisions on peers' investment behavior.

## 2. Hypothesis Development

### 2.1. Enterprise Practice of Peer Learning Theory

Relevant research based on social interaction theory shows that in order to avoid the risk problems caused by its information limitations and limited resources in the decision-making process, enterprises often choose to learn and follow the behaviors of other organizations with similar characteristics [40,41]. Graham and Harvey (2001) [42] surveyed

392 chief financial officers, and found that enterprises are not completely independent in making financial decisions and often interact socially. For example, the successful listing of enterprises will stimulate the willingness of peers to raise funds in the capital market. The practice of multinational corporations' social responsibility will drive the peers to fulfill more social responsibilities [43]. In addition, the company's capital structure [44], executive compensation [34], stock split [30], cash holding [45], dividend policy [36], M&A decision [37], investment decision [38,39], and other behaviors may be learned and imitated by peers, thus affecting the financial behavior of peer companies. As a new financing tool with both "green" and "financial" characteristics, green bonds can effectively solve the financing problem of green projects of enterprises [6,10], and, at the same time, it can help enterprises to establish a "green development" social image [2] and enhance the ESG value of enterprises. As the "pioneer" of the green bond market, the successful practice of green-bond-issuing enterprises will surely be learned and referenced by other enterprises, thus affecting the financial decisions of peer firms.

### 2.2. The Trigger Mechanism of Industry Green Innovation Spillover Effect by Green Bond

As a forward-looking strategy to fundamentally solve environmental problems, green innovation can help enterprises establish technical barriers and long-term competitive advantages by producing the dual value effects of the environment and finance. It is an important financial investment decision for enterprises [46]. Green innovation has natural development bottlenecks, such as high risk, long cycle, and significant investment [47], which creates fuzzy and difficult conditions for enterprises to implement green innovation decision making [48]. Previous studies have shown that enterprises' green technology innovation decisions are not only influenced by their organizational characteristics, resource conditions, and other factors but also closely related to the financial behavior of other firms [49]. Because peer enterprises are faced with the similar market environment and development prospects, when the pioneers of green bonds appear in the industry, it will inevitably lead to the learning and imitation of peer companies. Peer firms may adjust their green innovation level out of the instinct of "seeking benefits" and "avoiding harm", thus triggering the industry green innovation spillover effect.

### 2.2.1. Peer Firm's "Profit-Seeking" Motivation

First, the imitation effect. The successful practice of green bonds in enterprises has opened up a brand-new financing channel for peer enterprises and stimulated the willingness of peers to finance through the green bond market. Green bonds can not only play their financing function, providing a reliable long-term funding source for the development of green projects of enterprises, but also play a signal effect to establish the image of green and sustainable development of enterprises [2,15], thus guiding green government capital and green credit capital [50,51], as well as green social capital flowing into enterprises [13,14,16] to ease corporate financing constraints and reduce capital costs [9,12], optimizing the debt maturity structure. The successful experience of issuing green bonds by enterprises will be circulated in the industry through information channels, such as the media and the network of the board of directors. After learning the experience and benefits of green bond issuing enterprises, peer firms may make more strategic decisions that are beneficial to environmental protection, such as green innovation, so as to accumulate results for issuing green bonds in the future.

Second, the technology spillover effect. Green bonds can promote issuers to implement more green innovation activities and improve the industry's basic level of green technology. Zhang et al. (2022) [52] found that green bonds can improve the issuers' green innovation capability by easing financing constraints and improving information transparency; Wang and Feng (2022) [53] also found that in the green bond market in China, green bonds can significantly improve the green innovation level of issuers. The innovative products of green bond issuers can be used by peer firms for reference and transformation, thus

improving the green innovation efficiency of peers and then driving the iterative upgrading of new technologies in the industry.

### 2.2.2. Peer Firm's "Harm-Avoiding" Motivation

First, competitive pressure. The inherent green attribute of green bonds can help enterprises gain more investors' attention and green premium and enhance their financing advantages. Investing the funds from green bonds in green projects will improve the green performance of products, thus boosting consumers' demands for the green value of products. Faced with the change in the competitive environment, the interaction between them will make the peer companies respond to the green bond behavior of competitors to prevent the establishment of competition barriers and the loss of their advantages. According to the dynamic competition theory, innovation is the key for enterprises to shape their core competitiveness, while green innovation is critical for enterprises to seize the future market and achieve sustainable development [48]. Therefore, when the pioneering enterprises of green bonds appear in the industry, peers will pay more attention to the positive contribution of green technology innovation to financial performance and environmental performance and implement more green innovation activities.

Second, compliance pressure. According to the planning theory, individual behavior is influenced by the perception of stress during decision making [54], while the behaviors of other individuals in the reference group and the expectations of critical stakeholders for individuals constitute the primary sources of stress [55,56]. After green bonds guide green funds to flow into the industry, it will drive the orderly development of the green industry and simultaneously raise the cognitive threshold of the environmental legality of peer companies, resulting in subjective normative pressure and the willingness of enterprises to innovate green technologies. In addition, when an enterprise issues green bonds, it may indeed have a point-to-point effect, which will drive the environmental performance of the same group of enterprises to improve, thus triggering the environmental concern of the regulatory authorities and raising the threshold of environmental compliance, while standardizing in terms of weight will increase the environmental risks and compliance costs of enterprises. Therefore, for the motivation of crisis prevention, when peer enterprises observe that other enterprises in the industry successfully issue green bonds, they will improve their green innovation level to reduce uncertainty in the future.

To sum up, after learning the experience and benefits of issuing green bonds, the peer companies will adjust their level of green technology innovation for "seeking benefits" and "avoiding harm" and form the green innovation spillover effect of green bonds, the trigger mechanism of which is shown in Figure 1. Based on this, this paper proposes the following hypothesis:

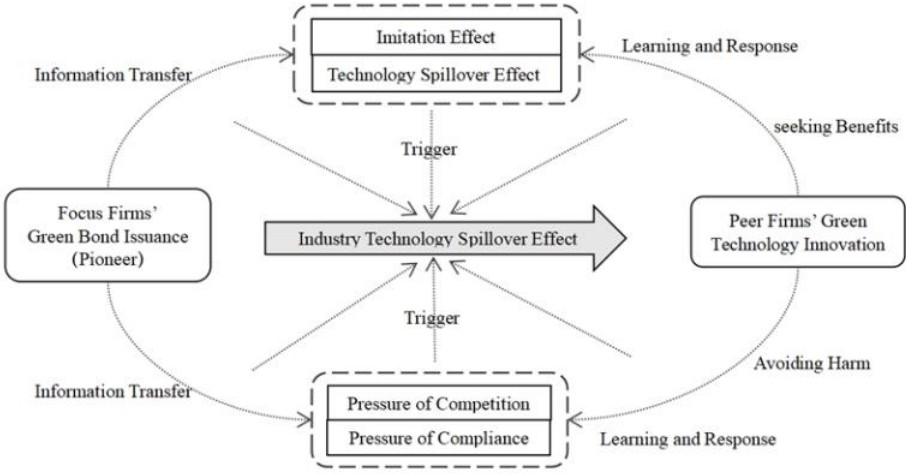

**Figure 1.** The trigger mechanism of industry green technology spillover effect by green bond.

**Hypothesis 1 (H1).** *Green bonds have an industry green technology spillover effect, and the issuance of corporate green bonds will promote the level of peer firms' green innovation.*

## 3. Research Methodology

### 3.1. Data Source and Sample

The Wind database comprehensively includes the announcement date, circulation, issue price, basic information of issuers, and other data on green bonds issued by China enterprises. The information on the green bonds database has been updated to July 2022. However, we were only able to obtain the green innovation data before 2020, so the data analyzed in this paper are relevant up to the end of 2020. A total of 1324 green bonds were collected in this paper. As it is impossible to observe which green projects government bonds and financial bonds are invested in, the bonds issued by financial enterprises or local governments were excluded from the sample; thus, 1001 green bonds were obtained, and the issuers involved 38 industries. Although the target company can learn from different types of companies, due to the similarity of factors, such as industry environment and market competition, and the consensus formed by closed circles, the target company is more inclined to interact with companies in the same industry. Therefore, we took the listed companies that had not issued green bonds in these 38 industries as the treatment group (2245 in total) and the listed companies in other industries as the control group (2690 in total) and observed the impact of the green bonds on the green innovation behavior of peers. Based on the data of non-financial listed companies in Shanghai and Shenzhen A-shares in China from 2010 to 2020, this paper used PSM one-to-one nearest neighbor matching method to match the appropriate control companies for the treatment group and excluded the ST companies and the samples with missing required indicators, and, finally, we obtained 3606 sample companies, with a total of 24,858 firm-year observations. Green innovation data was obtained from the Chinese Research Data Services (CNRDS) database. CNRDS is a high-quality, open, and platform-based comprehensive data platform for Chinese economic, financial, and business research and is a very commonly used and reliable database for Chinese academic research. The other related data was obtained from the China Stock Market & Accounting Research Database (CSMAR). CSMAR database is a professional economic and financial data platform. The database covers a number of research series, including China stock market, China listed companies, China fund market, China bond market, China derivatives, China economy, China money market, special research, etc., including structured financial statements, trading quotes, unstructured news information, research reports, company announcement data, etc. It is one of the most comprehensive economic and financial research databases in China at present. When we found suspicious data, we checked the company's annual report. For example, we found that the return on equity of Qinghai Salt Lake Industry Co., Ltd. was −19.28% and −19.16% in 2017 and 2018, respectively, and it suddenly became 160.52% in 2019. This phenomenon is abnormal, so we checked the annual report to find out the reason. We checked more than 200 pieces of data, accounting for about 1% of the total sample. We did not find that the sample data were inconsistent with the annual report data, which also made us more convinced that the data in the database we adopted were reliable. In addition, in order to reduce the influence of extreme values, all continuous variables were treated with 1% and 99% quantile winsorized.

### 3.2. Research Model and Variable Definition

#### 3.2.1. Research Model

Due to the different time points for enterprises to issue green bonds, referring to [3,4,57], a gradual double difference model was set:

$$Patent_{i,t} = \alpha_0 + \alpha_1 Green_i \times Post_t + \sum \alpha_k Controls_{i,t} + \lambda_i + \delta_t + \varepsilon_{i,t} \tag{1}$$

$Patent_{i,t}$ was the explained variable of this paper, which indicated the level of green innovation of enterprise $i$ in $t$ year. We measured it from the quantity and quality of green innovation. $Green_i \times Post_t$ was a double difference term in the DID model. This paper focused on the coefficient $\alpha_1$ of $Green_i \times Post_t$. If $\alpha_1$ was significantly positive, it meant that the issuance of corporate green bonds could promote the green innovation level of peers. $Controls_{i,t}$ was the control variable group of the model. In addition, Model 1 also controlled the firm individual fixed effect $\lambda_i$ and annual fixed effect $\delta_t$.

### 3.2.2. Variable Definition

(1)    Explained variable

Peer firms' green innovation ($Patent_{i,t}$) was the explained variable. Green innovation refers to the enterprise's green product design and process innovation, as well as organizational management support and innovative implementation, involving energy conservation, pollution prevention, waste recycling, and other aspects that deal with environmental problems and achieve specific environmental protection goals and sustainable development. In the existing literature, some scholars use the sales revenue of new products per unit energy consumption to measure the level of green technology innovation of enterprises [58]. Although it can reflect the degree of green innovation of enterprises to some extent, it is difficult to describe the green R&D achievements of enterprises. In contrast, green patent requires enterprises to research, develop, popularize, and apply green technology, which is the dominant output of green technology innovation of enterprises, and thus can better reflect the green innovation capability of enterprises [27,59]; this paper used the research methods of Lin and Ma (2022) [32] and Xia et al. (2022) [25] for reference and identified the green patents applied by enterprises, according to IPC Green Inventory issued by the World Intellectual Property Organization (WIPO). The steps are as follows: (1) Obtain the IPC Green Inventory published by the WIPO in 2010, a convenient retrieval tool for green patents launched. According to the IPC Green Inventory, green patents include seven fields: waste management, nuclear power, transportation, energy conservation, agriculture and forestry, alternative energy production, and administrative supervision and design, involving about 200 areas that are directly friendly to the environment. (2) Obtain the patent information of China A and share listed companies from the CNRDS database. CNRDS provides information on the number, classification, and citation of patents applied by Chinese companies. (3) Match the IPC Green Inventory with the patent information of listed companies in China, according to the IPC number, to obtain the listed companies' green-patent-related data. In China, patents are divided into invention patents, utility model patents, and design patents. An invention patent refers to a new technical proposal for a product, method, or improvement. A utility model patent refers to a new technical scheme suitable for practical use, based on product shape, structure, or combination. The granting of a utility model patent does not require substantial examination, so the procedure is relatively simple, and the cost is relatively low. A design patent refers to a new design rich in aesthetic feeling and suitable for industrial application by the shape, pattern, or combination of products and the combination of color, shape, and pattern. Compared with utility model patents and design patents, invention patents have a relatively high degree of innovation, a longer development cycle, more significant difficulty in research and development, and more complicated application procedures. Following [60,61], we used the number of green invention patent applications to characterize the quality of green innovation. In addition, the number of green patent citations can also explain the patent quality well. Therefore, we used three indicators to measure green innovation. (1) The total number of green patent applications (Patent1) refers to the number of green patents applied by enterprises in a fiscal year. When precisely calculated, we added 1 to the total number of green patent applications in that year and then took the natural logarithm. This indicator represented the quantity of green innovations. (2) The number of green invention patent applications (Patent2) refers to the total number of green invention patents applied by enterprises in a fiscal year. When calculating precisely, we added 1 to the number of

invention green patents applied for that year and then took the natural logarithm. This indicator represented the quality of green innovation. (3) The amount of green patents cited (Patent3) refers to the sum of the number of patents cited by an enterprise in a fiscal year. In the specific calculation, we added 1 to the number of green patents cited in that year and then took the natural logarithm. This indicator also represented the quality of green innovation.

(2)    Explanatory variables

$Green_i \times Post_t$ was the core explanatory variable in this paper. $Green_i$ was the dummy variable of the treatment group and the control group. If an enterprise in a specific industry publicly issued green bonds, other enterprises in the same industry will be assigned a value of 1, otherwise, it will be assigned a value of 0. The same industry adopts the fourth industry classification of Wind industry classification. Wind industry classification standard is based on the Global Industry Classification System (GICS) and adopts a four-level industry classification system, including 11 first-level industries, 24 s-level industries, 62 third-level industries, and 136 fourth-level industries. It is a common industry classification standard in China. $Post_t$ was a virtual variable of time. For the treatment group, if the issuing time of the first green bond enterprise in an industry was $t$, the enterprises in the industry were assigned a value of 1 at time $t$ and later, otherwise they were assigned a value of 0; For the control group, all $Post_t$ values were 0.

(3)    Control variables

Many factors drive the level of green technology innovation of enterprises, so we included several groups of essential control variables in the model. Firstly, based on the resource-based theory, enterprises' willingness and risk-taking level to innovate are influenced by their characteristics and capabilities. Therefore, we controlled the variables of enterprise size (*Size*), return on net assets (*Roe*), capital expenditure (*Expend*), cash flow (*Cfo*), equity nature (*Soe*), and enterprise age (*Age*) in the model. Secondly, based on the stakeholder theory, the demands and governance ability of stakeholders, such as creditors and enterprise executives, also interfere with the enterprises' green technology innovation decisions. Therefore, the asset–liability ratio (*Lev*) and executive shareholding ratio (*Exe*) were included in the control variable group. Thirdly, because of the externality of environmental governance and technology, enterprises often lack the original motivation of green technology innovation, so government intervention plays an important role in the decision making of green technology innovation. Based on this, this paper controlled two factors: government subsidy (*Subsidy*) and environmental regulation (*Regula*) in the model. See Table 1 for definitions of the main variables.

**Table 1.** Variables description.

| Variables | Description |
|---|---|
| Patent1 | Add 1 to the total number of green patent applications in that year and then take the natural logarithm. |
| Patent2 | Add 1 to the number of invention green patents applied for in that year and then take the natural logarithm. |
| Patent3 | Add 1 to the number of green patents cited in that year and then take the natural logarithm. |
| Green × Post | If an enterprise issues green bonds in t year, other enterprises in the same industry will be assigned a value of 1 in t year and later, otherwise it will be assigned a value of 0. |
| Size | Natural logarithm of total assets at the end of the fiscal year. |
| Roe | Annual net profit divided by total owner's equity at the end of fiscal year. |
| Expend | Cash paid for the purchase and construction of fixed assets, intangible assets, and other long-term assets divided by total assets at the end of fiscal year. |
| Cfo | Net cash flow from operating activities divided by total assets at the end of fiscal year. |
| Soe | If the enterprise is state-controlled, take one, otherwise take zero. |

**Table 1.** *Cont.*

| Variables | Description |
|---|---|
| Age | Time interval between enterprise registration year and enterprise sample year. |
| Lev | Liabilities divided by total assets at the end of fiscal year. |
| Exe | Number of shares held by executives divided by total share capital at the end of the fiscal year. |
| Subsidy | Total government subsidies received by enterprises divided by operating income. |
| Regula | The investment amount of pollution control in each province is divided by the total industrial output value of the province and multiplied by 1000. |

## 4. Empirical Results

### 4.1. Descriptive Statistical

Table 2 provides descriptive statistical results of major variables. The standard deviations of total green patent applications (*Patent1*), green invention patent applications (*Patent2*), and green patents cited (*Patent3*) are 1.105, 0.897, and 1.597, respectively, which indicates that there are significant differences in the level of green innovation among enterprises. We tested the average difference before and after the impact of green bonds by industry in the treatment group. Table 3 reports the industry results that passed the test of at least 10% confidence level difference. We sorted them according to the average difference of the total number of green patent applications (Patent1). It can be seen that when an enterprise in an industry issues green bonds, the total number of green patent applications (Patent1), green invention patent applications (Patent2), and green patent citations (Patent3) of other enterprises in most industries are significantly increased. In addition, we also found that the green innovation level of comprehensive industries has the most significant effect, followed by environmental and facility services, while the green innovation level of gold, automobile manufacturing, seaport and service industries, and airport service industries has declined instead of rising. These industries should be further studied and paid attention to by scholars and government departments.

**Table 2.** Descriptive statistics.

| Variable | Obs | Mean | Median | S.D. | Min | Max |
|---|---|---|---|---|---|---|
| Patent1 | 24,858 | 0.811 | 0 | 1.105 | 0 | 4.331 |
| Patent2 | 24,858 | 0.541 | 0 | 0.897 | 0 | 3.871 |
| Patent3 | 24,858 | 1.075 | 0 | 1.597 | 0 | 6.068 |
| Green | 24,858 | 0.500 | 1 | 0.500 | 0 | 1 |
| Size | 24,858 | 22.020 | 21.890 | 1.185 | 19.710 | 25.540 |
| Roe | 24,858 | 0.059 | 0.070 | 0.133 | −0.766 | 0.335 |
| Lev | 24,858 | 0.408 | 0.401 | 0.198 | 0.052 | 0.895 |
| Expend | 24,858 | 0.051 | 0.037 | 0.047 | 0 | 0.228 |
| Cfo | 24,858 | 0.048 | 0.047 | 0.066 | −0.144 | 0.234 |
| Soe | 24,858 | 0.346 | 0 | 0.476 | 0 | 1 |
| Age | 24,858 | 17.030 | 17 | 5.798 | 4 | 32 |
| Subsidy | 24,858 | 10.430 | 15.410 | 8.053 | 0 | 19.970 |
| Exe | 24,858 | 0.077 | 0.001 | 0.144 | 0 | 0.617 |
| Regula | 24,858 | 22.541 | 35.671 | 16.832 | 1.679 | 269.876 |

We further test the correlation of the research variables, and the results are shown in Table 4: First, green bonds are significantly positively correlated with the total number of green patent applications (*Patent1*), green invention patent applications (*Patent2*), and green patent citations (*Patent3*), hypothesis 1 is preliminarily verified. Secondly, the control variables, such as enterprise size (*Size*), return on net assets (*Roe*), capital expenditure (*Expend*), cash flow (*Cfo*), equity nature (*Soe*), enterprise age (*Age*), government subsidy (Subsidy), the asset–liability ratio (*Lev*), executive shareholding ratio (Exe), and environmental regulation

(Regula), are related to green innovation (*Patent1/Patent2/Patent3*). Therefore, it is reasonable to bring these variables into Model 1. In addition, the correlation coefficients among the control variables are small, which means that there is no obvious multicollinearity problem among the main variables selected in this paper. Further, we test Model 1 by VIF one by one, and the test results show that the maximum VIF value is 3.56, which is far less than 10, indicating that Model 1 is less affected by multicollinearity and its estimation result is reliable.

**Table 3.** Difference of average value of green innovation before and after the impact of green bonds in various industries.

| Industry | Patent1 | Patent2 | Patent3 | Industry | Patent1 | Patent2 | Patent3 |
|---|---|---|---|---|---|---|---|
| Comprehensive industry | −1.246 | −0.686 | −0.802 | Diversified chemical industry | −0.398 | −0.301 | −0.043 |
| Environmental and facility services | −0.96 | −0.766 | −0.949 | Basic chemical industry | −0.392 | −0.227 | −0.309 |
| Architecture and engineering | −0.920 | −0.654 | −0.518 | Electrical components and equipment | −0.338 | −0.346 | −0.249 |
| Building materials | −0.679 | −0.523 | −0.342 | Refining and sales of oil and natural gas | −0.337 | 0.021 | −0.146 |
| Water | −0.615 | −0.443 | −0.668 | Heavy electrical equipment | −0.256 | −0.214 | −0.223 |
| Chemical fiber | −0.535 | −0.342 | −0.168 | Leisure facilities | −0.256 | −0.167 | −0.187 |
| Electric power | −0.531 | −0.376 | −0.468 | Coal and consumer fuel | −0.219 | −0.152 | −0.080 |
| Semiconductor products | −0.489 | −0.246 | −0.258 | Trading companies and distributors of industrial products | −0.190 | −0.148 | −0.353 |
| Steel | −0.483 | −0.472 | −0.380 | Gold | −0.185 | −0.125 | 0.064 |
| Metal nonmetal | −0.478 | −0.238 | −0.133 | Automobile industry | −0.113 | −0.123 | 0.125 |
| Highway and railway | −0.408 | −0.262 | −0.415 | Seaports and services | −0.108 | 0.098 | −0.013 |
| New energy generator | −0.403 | −0.282 | −0.429 | Airport service | 0.124 | 0.108 | 0.025 |
| Real estate development | −0.401 | −0.237 | −0.063 | | | | |

*4.2. Propensity Score Matching Results*

4.2.1. Balance Test

If an enterprise in an industry successfully issues green bonds, other enterprises in the industry that have not issued green bonds are the treatment group, while enterprises in other industries are the control group. For matching objects, as learned from HAO et al. (2018) [62], the indicators of the treatment group enterprises were selected in the year before the policy impact to match the data of the control group enterprises in the same period. For example, if an enterprise in an industry successfully issues green bonds for the first time in 2017, and the enterprise is the first enterprise in the industry to successfully issue green bonds, the industry enterprises will use the indicators from 2016 to match the data of the control group enterprises in 2016. In the specific matching, the tendency score is estimated by the logit model, and one-to-one non-put-back neighbor matching is adopted, and only individuals within the common value range are matched. Covariates include enterprise size (Size), return on net assets (Roe), asset–liability ratio (Lev), cash flow (Cfo), equity nature (Soe), and control of the industry.

**Table 4.** Correlation results.

| | Patent1 | Patent2 | Patent3 | Green × Post | Size | Roe | Lev | Expend | Cfo | Soe | Age | Subsidy | Exe |
|---|---|---|---|---|---|---|---|---|---|---|---|---|---|
| Patent1 | 1 | | | | | | | | | | | | |
| Patent2 | 0.919 *** | 1 | | | | | | | | | | | |
| Patent3 | 0.630 *** | 0.609 *** | 1 | | | | | | | | | | |
| Green×Post | 0.149 *** | 0.128 *** | 0.110 *** | 1 | | | | | | | | | |
| Size | 0.386 *** | 0.375 *** | 0.414 *** | 0.129 *** | 1 | | | | | | | | |
| Roe | 0.047 *** | 0.049 *** | −0.009 | −0.017 *** | 0.098 *** | 1 | | | | | | | |
| Lev | 0.183 *** | 0.160 *** | 0.215 *** | 0.098 *** | 0.440 *** | −0.169 *** | 1 | | | | | | |
| Expend | −0.009 | −0.014 ** | −0.052 *** | −0.047 *** | −0.024 *** | 0.111 *** | −0.050 *** | 1 | | | | | |
| Cfo | −0.007 | −0.001 | 0.004 | 0.027 *** | 0.143 *** | 0.281 *** | −0.079 *** | 0.132 *** | 1 | | | | |
| Soe | 0.053 *** | 0.064 *** | 0.135 *** | 0.002 | 0.332 *** | −0.012 * | 0.268 *** | −0.098 *** | 0.030 *** | 1 | | | |
| Age | 0.041 *** | 0.047 *** | 0.105 *** | 0.204 *** | 0.183 *** | −0.059 *** | 0.171 *** | −0.187 *** | 0.042 *** | 0.172 *** | 1 | | |
| Subsidy | 0.222 *** | 0.209 *** | 0.172 *** | 0.342 *** | 0.210 *** | −0.030 *** | −0.016 ** | −0.135 *** | 0.090 *** | −0.112 *** | 0.336 *** | 1 | |
| Exe | −0.036 *** | −0.033 *** | −0.128 *** | −0.001 | −0.278 *** | 0.062 *** | −0.263 *** | 0.111 *** | −0.021 *** | −0.369 *** | −0.197 *** | 0.051 *** | 1 |
| Regula | 0.032 *** | 0.027 *** | −0.013 | 0.056 | 0.134 | 0.245 | −0.312 | 0.215 * | 0.001 | 0.078 | −0.12 | 0.278 *** | 0.110 |

Notes: *, **, *** Significant at 10, 5, and 1% levels, respectively; *t* values are reported in parentheses; refer to Table 1 for variables description.

The premise of PSM effectiveness is that there is no significant difference in observable variables between the matched treatment group and the control group. Therefore, the matching balance test is carried out in this paper, and the results are shown in Table 5. First, compared with the variables before matching (unmatched), the standardized deviation (%bias) of all matched variables (matched) is greatly reduced. Second, the standardized deviation (%bias) of all the matched variables (matched) is less than 10%. Third, the *t*-test results of all matched variables (matched) accept the original assumption that there is no systematic difference between the treatment and control groups. The above results show that the observable variables selected in this paper and the matching methods are appropriate. Figure 2 shows the kernel density functions before and after matching. It can be seen that the deviation of the two kernel density curves before matching is significant, and the two curves are closer due to the narrowing of the mean distance after matching, which shows that the matching is effective and reasonable to some extent. In this way, the estimation deviation caused by the "self-selection problem" of samples can be better solved.

**Table 5.** PSM balanced test.

| Variable | Unmatched Matched | Mean | | %Bias | %Reduct \|bias\| | *t*-Test | |
|---|---|---|---|---|---|---|---|
| | | Treated | Control | | | t | *p* > \|t\| |
| Size | Unmatched | 22.58 | 21.996 | 48.4 | | 26.64 | 0.000 |
| | matched | 22.573 | 22.55 | 1.9 | 96.1 | 0.87 | 0.387 |
| Roe | Unmatched | 0.0463 | 0.0578 | −7.7 | | −4.24 | 0.000 |
| | matched | 0.0467 | 0.0462 | 0.4 | 95.1 | 0.18 | 0.861 |
| Lev | Unmatched | 0.4311 | 0.3833 | 25.2 | | 13.72 | 0.000 |
| | matched | 0.4307 | 0.4300 | 0.3 | 98.6 | 0.17 | 0.868 |
| Cfo | Unmatched | 0.0522 | 0.0536 | −2.1 | | −1.15 | 0.248 |
| | matched | 0.0523 | 0.0517 | 0.9 | 56.3 | 0.46 | 0.646 |
| Soe | Unmatched | 0.3901 | 0.1973 | 43.3 | | 23.88 | 0.000 |
| | matched | 0.3887 | 0.3779 | 2.4 | 94.4 | 1.09 | 0.277 |

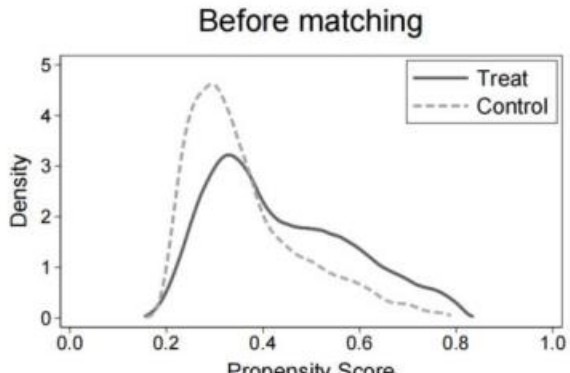

**Figure 2.** Kernel density function plot before and after matching.

### 4.2.2. Average Treatment Effect

Table 6 reports the average processing effect of PSM, in which "Unmatched" reports the estimated results of the samples before matching. The differences between the treatment group and the control group before matching are 0.4902, 0.3316, and 0.6440, respectively, all of which are significant at the 1% level; "ATT" is the average treatment effect of the participants after PSM. The differences between the treatment group and the control group are 0.4768, 0.3226, and 0.6280, respectively, and they are all significant at the level of 1%. The above results support hypothesis 1: the issuance of corporate green bonds can significantly improve the peer firms' green technology innovation.

**Table 6.** PSM average treatment effect.

| Variable | Sample | Treatment Group | Control Group | Difference | S.D. | *t* Test |
|---|---|---|---|---|---|---|
| *Patent1* | Unmatched | 1.0384 | 0.5482 | 0.4902 | 0.0128 | 38.42 |
| | ATT | 1.0387 | 0.5618 | 0.4768 | 0.0131 | 36.44 |
| | ATU | 0.5482 | 1.0145 | 0.4663 | | |
| | ATE | | | 0.4712 | | |
| *Patent2* | Unmatched | 0.6962 | 0.3646 | 0.3316 | 0.0105 | 31.55 |
| | ATT | 0.6964 | 0.3738 | 0.3226 | 0.0107 | 29.89 |
| | ATU | 0.3646 | 0.6732 | 0.3086 | | |
| | ATE | | | 0.3150 | | |
| *Patent3* | Unmatched | 1.3834 | 0.7394 | 0.6440 | 0.0185 | 34.68 |
| | ATT | 1.3837 | 0.7557 | 0.6280 | 0.0190 | 33.05 |
| | ATU | 0.7394 | 1.3491 | 0.6097 | | |
| | ATE | | | 0.6182 | | |

### *4.3. Regression Results*

#### 4.3.1. Baseline Regression Results

This paper performs the Model 1 test based on PSM processing. Table 7, respectively, reports the impact of the issuance of corporate green bonds on the total number of green patent applications (*Patent1*), green invention patent applications (*Patent2*), and green patent citations (*Patent3*) of peer enterprises. Columns 1 to 3 report the regression results without added control variables. It can be seen that the coefficients of *Green* × *Post* are 0.105, 0.069, and 0.145, respectively, and they are all significant at the 1% level. By adding control variables to the model, the results are shown in columns 4 to 6 of Table 5, and the coefficient of *Green* × *Post* is still significantly positive at 1%. The above results show that the issuance of green bonds can significantly promote the quantity and quality of peer firms' green patents. Hypothesis 1 has been verified.

**Table 7.** PSM-DID test results.

| Variable | (1) | (2) | (3) | (4) | (5) | (6) |
|---|---|---|---|---|---|---|
| | *Patent1* | *Patent2* | *Patent3* | *Patent1* | *Patent2* | *Patent3* |
| Green × Post | 0.105 *** | 0.069 *** | 0.145 *** | 0.099 *** | 0.065 *** | 0.136 *** |
| | (3.83) | (3.00) | (4.15) | (3.84) | (2.95) | (4.01) |
| Size | | | | 0.324 *** | 0.263 *** | 0.210 *** |
| | | | | (13.84) | (12.68) | (7.73) |
| Roe | | | | 0.013 | 0.006 | −0.173 *** |
| | | | | (0.29) | (0.16) | (−2.84) |
| Lev | | | | 0.065 | 0.040 | 0.499 *** |
| | | | | (0.91) | (0.68) | (5.37) |
| Expend | | | | 0.447 *** | 0.423 *** | 0.075 |
| | | | | (2.64) | (3.05) | (0.34) |
| Cfo | | | | −0.171* | −0.114 | 0.282 ** |
| | | | | (−1.85) | (−1.49) | (2.28) |
| Soe | | | | 0.057 | 0.058 | 0.124 * |
| | | | | (0.99) | (1.26) | (1.81) |
| Age | | | | −0.036 | −0.028 | 0.003 |
| | | | | (−1.48) | (−1.27) | (0.07) |
| Subsidy | | | | 0.020 *** | 0.019 *** | 0.034 *** |
| | | | | (8.01) | (8.57) | (10.95) |
| Exe | | | | −0.048 | 0.050 | −0.551 *** |
| | | | | (−0.47) | (0.60) | (−3.95) |
| Regula | | | | 0.011 *** | 0.009 ** | −0.107 * |
| | | | | (2.98) | (2.17) | (−1.85) |
| Constant | 0.331 *** | 0.181 *** | −0.018 | −6.226 *** | −5.167 *** | −4.696 *** |
| | (17.01) | (11.24) | (−0.54) | (−10.77) | (−10.00) | (−6.54) |
| Year and firm fixed effect | YES | YES | YES | YES | YES | YES |
| Observations | 24,858 | 24,858 | 24,858 | 24,858 | 24,858 | 24,858 |
| adj_$R^2$ | 0.653 | 0.628 | 0.747 | 0.670 | 0.646 | 0.755 |
| F | 110.1 | 69.98 | 132.9 | 73.39 | 47.42 | 83.28 |

Notes: *, **, *** Significant at 10, 5, and 1% levels, respectively; *t* values are reported in parentheses; refer to Table 1 for variables description.

### 4.3.2. Parallel Trend and Dynamic Effect Test

The double difference model needs to satisfy the parallel trend hypothesis, that is, the development trend of the explained variables in the treatment group and the control group is the same without policy intervention. We build Model 2 to test the parallel trend of the progressive DID model and further analyze the dynamic marginal effect of the green bonds on the green innovation of enterprises. In Model 2, *Before(>3)/Before(3)/Before(2)/Before(1)/ Current/After(1)/After(2)/After(>2)*, respectively, represent the interaction items of $Green_i \times Post_{<(t-3)}/Green_i \times Post_{t-3}/Green_i \times Post_{t-2}/Green_i \times Post_{t-1}/Green_i \times Post_t/Green_i \times Post_{t+1}/Green_i \times Post_{t+2}/Green_i \times Post_{>(t+2)}$.

$$Patent_{i,t} = \alpha_0 + \alpha_1 Before(>3) + \alpha_2 Before(3) + \alpha_3 Before(2) + \alpha_4 Before(1) + \alpha_5 Current + \alpha_6 After(1) + \alpha_7 After(2) + \alpha_8 After(>2) + \sum \alpha_j Controls_{i,t} + \varepsilon_{i,t} \quad (2)$$

Table 8 reports the parallel trend and dynamic effect test results. It can be seen that the regression coefficients of *Before(3)*, *Before(2)*, and *Before(1)* are not significant, irrespective of the explained variables of the total green patent applications (*Patent1*), green invention patent applications (*Patent2*) or green patent citations (*Patent3*), indicating that there is no significant difference in the green innovation level between the enterprises in the treatment group and those in the control group before the issuance of green bonds. In addition, the coefficients of *Current*, *After(1)*, and *After(2)* are basically significantly positive, indicating that issuing green bonds can continuously promote the green innovation level of peer enterprises.

**Table 8.** Parallel trend and dynamic effect test results.

| Variable | (1) | (2) | (3) |
|---|---|---|---|
| | *Patent1* | *Patent2* | *Patent3* |
| Before (3) | −0.033 (−0.73) | −0.043 (−1.11) | 0.200 (0.82) |
| Before (2) | 0.026 (0.61) | −0.007 (−0.20) | 0.252 (1.17) |
| Before (1) | 0.066 (1.64) | −0.002 (−0.06) | 0.259 * (1.86) |
| Current | 0.319 *** (11.49) | 0.219 *** (9.56) | 0.470 *** (11.70) |
| After (1) | 0.438 *** (13.87) | 0.228 *** (8.74) | 0.415 *** (9.10) |
| After (2) | 0.408 *** (12.96) | 0.263 *** (10.12) | 0.391 *** (8.58) |
| Controls | YES | YES | YES |
| Constant | −6.941 *** (−51.17) | −5.676 *** (−50.62) | −10.586 *** (−53.95) |
| Observations | 24,858 | 24,858 | 24,858 |
| adj_R$^2$ | 0.220 | 0.192 | 0.219 |

**Notes**: *, *** Significant at 10, and 1% levels, respectively; *t* values are reported in parentheses; refer to Table 1 for variables description.

### 4.3.3. Robustness Test

(1)    Replace the PSM matching method

In order to alleviate the research result error caused by the loss of one-to-one matching samples, this paper adopts the kernel matching method to determine the weight again, applies the condition of "common support", and finally obtains 28,805 firm-year observations. W re-estimate Model 1 with new samples, which are shown in Table 9 columns 1–3. No matter what the explained variables of the total green patent applications (*Patent1*), green invention patent applications (*Patent2*), and green patent citations (*Patent3*) are, the coefficient of *Green×Post* is always positive and significant at the 1% confidence level. Hypothesis 1 is verified again, which shows that the results of this paper are robust.

**Table 9.** Robustness test.

| | (1) | (2) | (3) | (4) | (5) | (6) | (7) | (8) | (9) |
|---|---|---|---|---|---|---|---|---|---|
| Variable | Replace the PSM Matching Method | | | Tobit Model | | | Reduce the Influence of Industry Policies | | |
| | *Patent1* | *Patent2* | *Patent3* | *Patent1* | *Patent2* | *Patent3* | *Patent1* | *Patent2* | *Patent3* |
| Green × Post | 0.122 *** | 0.087 *** | 0.183 *** | 0.294 *** | 0.211 *** | 0.180 *** | 0.167 *** | 0.097 *** | 0.112 *** |
| | (4.70) | (3.91) | (5.52) | (9.49) | (6.50) | (3.17) | (10.32) | (7.26) | (4.76) |
| Controls | YES | YES | YES | YES | YES | YES | YES | YES | YES |
| Constant | −4.998 *** | −4.199 *** | −4.215 *** | −14.121 *** | −14.897 *** | −25.596 *** | −6.803 *** | −5.584 *** | −10.379 *** |
| | (−11.67) | (−11.21) | (−9.13) | (−49.53) | (−48.59) | (−52.48) | (−46.27) | (−45.25) | (−51.87) |
| Observations | 28,805 | 28,805 | 28,805 | 24,858 | 24,858 | 24,858 | 21,356 | 21,356 | 21,356 |
| adj_R$^2$ | 0.671 | 0.647 | 0.759 | | | | 0.571 | 0.609 | 0.687 |
| Pseudo R$^2$ | | | | 0.068 | 0.074 | 0.063 | | | |

**Notes**: *** Significant at 1% level; *t* values are reported in parentheses; refer to Table 1 for variables description.

(2)    Tobit model

As the explained variables of this paper, total green patent applications (*Patent1*), green invention patent applications (*Patent2*), and green patent citations (*Patent3*) are all truncated variables with zero as the lower limit, which may lead to errors in OLS estimation results; therefore, the Tobit model is used again in this paper for estimation, and the results are shown in Table 9 columns 4–6. The coefficients of *Green × Post* are 0.294, 0.211, and 0.180, respectively, all of which are significant at the level of 1%, which indicates that the issuance of green bonds by enterprises can indeed improve the green innovation level of peer enterprises and also supports hypothesis 1.

(3)    Reduce the influence of industry policies

Although this paper alleviates the influence of industry policies by controlling the fixed effect of firms, the green bond issuers in the sample are mainly distributed in three

industries, including the thermal power production and supply industry, the ecological protection and environmental management industry, and the civil engineering and construction industry, accounting for nearly half of all green bond issuers. In order to alleviate the possible impact of industry aggregation, this paper excludes the samples of bond issuers belonging to these three industries and carries out the PSM-DID test again. The regression results are shown in Table 9 columns 7–9. The coefficient of *Green × Post* is still significant, which shows that the results of this paper are robust.

(4)    Placebo test

In order to alleviate the influence of other unobservable factors on the research results, we constructed the counter-evidence by repeated sampling regression. Specifically, we first selected 2245 companies from the sample companies as treatment groups by random sampling method. Then, we used one-to-one non-put-back neighbor matching to match the appropriate control groups in other companies. Finally, we used the new samples to estimate Model 1. We repeated the extraction 1000 times, and the result is shown in Figure 3. The straight line is the T value estimated by the baseline regression model. Irrespective of the explained variable of the total number of green patent applications (*Patent1*), the number of green invention patent applications (*Patent2*), and the number of green patents cited (*Patent3*), the t values of most random sampling results are near zero, and only a few estimated results have t values larger than the baseline regression results. This means that the promotion effect of green bonds on green innovation of peer enterprises is not caused by other unobservable factors.

### 4.4. Heterogeneity Test

So far, our research shows that the successful practice of issuing green bonds will be learned by peer enterprises, thus promoting the level of green technology innovation of peers. So, we are interested in what kind of enterprises that issue green bonds are more likely to be learned by their peers.

### 4.4.1. Industry Leaders

Studies have found that the behavior of industry leaders easily influences the behavior of other enterprises in the same industry. For example, Brown et al. (2018) [63] found that when an industry leader receives a letter of inquiry about financial reports, the company that did not receive the letter will pay more attention to the publicly inquired opinions of the company that received the letter and improves its information disclosure quality. Bratten et al. (2016) [64] found that enterprises in the same industry will pay attention to the reported earnings of industry leaders. If the performance of industry leaders exceeds analysts' expectations, the motivation of other enterprises in the same industry to manipulate earnings will also be enhanced. Therefore, we believe that when industry leaders issue green bonds, other enterprises in the same industry will pay more attention to green bonds and green behaviors. Referring to Wu et al. (2022) [3], we define companies whose operating income accounts for more than 3% of industry income as industry leaders. If the green bond issuer is an industry leader, the variable *Leader* value is 1; otherwise, it is 0. We put *Leader* and *Green × Post × Leader* into Model 1 at the same time. The results are shown in Table 10 columns 1–3. The coefficients of *Green × Post × Leader* are all significantly positive at the 1% level, indicating that when the green bond issuer is an industry leader, it has a more decisive influence on the green innovation level of the peer company.

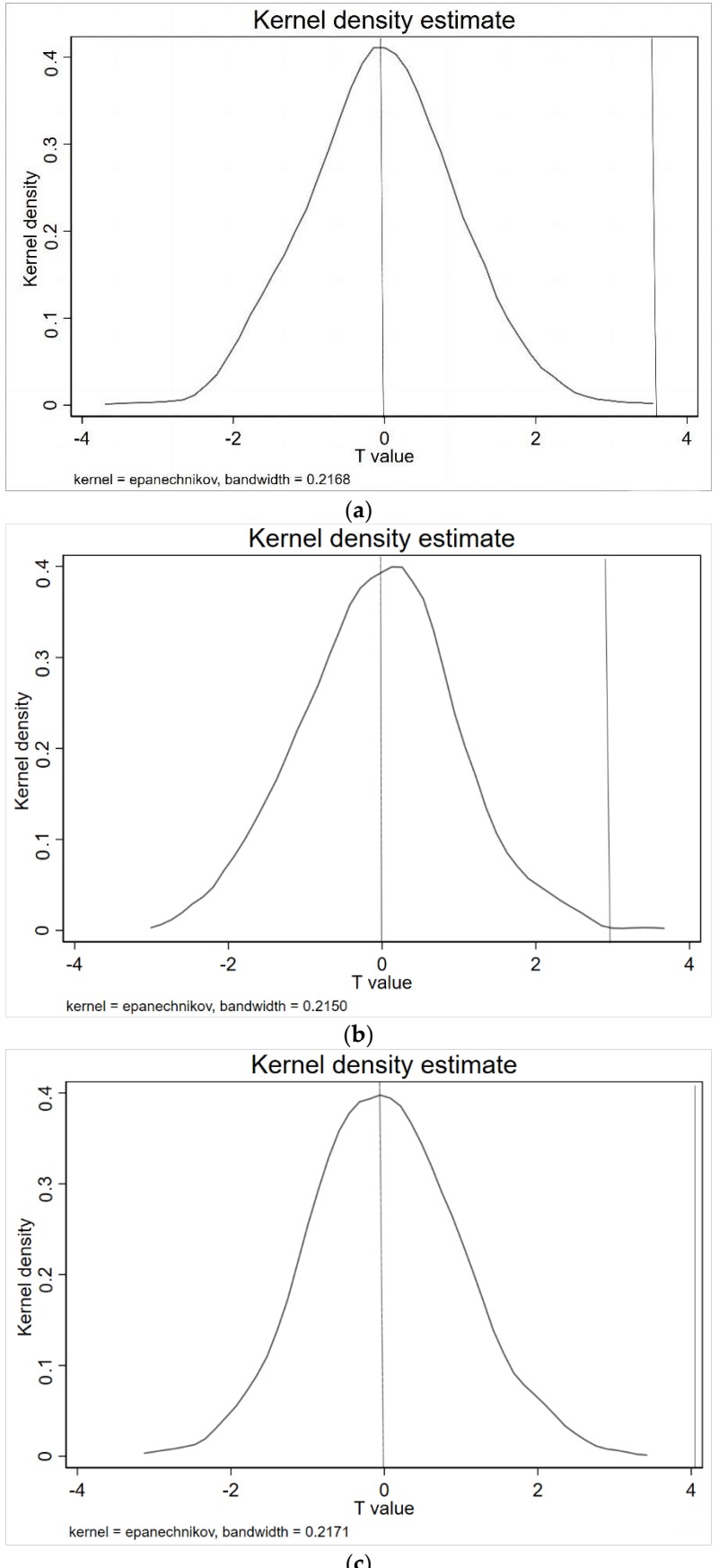

**Figure 3.** Placebo test results. (**a**) *Patent1*. (**b**) *Patent2*. (**c**) *Patent3*.

**Table 10.** Heterogeneity test results (a).

| Variable | (1) | (2) | (3) | (4) | (5) | (6) |
|---|---|---|---|---|---|---|
| | Industry Leaders | | | Issuer with High Media Attention | | |
| | *Patent1* | *Patent2* | *Patent3* | *Patent1* | *Patent2* | *Patent3* |
| Green × Post | 0.076 *** | 0.082 *** | 0.108 *** | 0.049 *** | 0.052 *** | 0.109 *** |
| | (4.18) | (4.97) | (4.05) | (2.99) | (4.36) | (4.02) |
| Leader | 0.232 *** | 0.308 *** | 0.277 *** | | | |
| | (3.89) | (3.09) | (4.23) | | | |
| Green × Post × Leader | 0.102 *** | 0.098 *** | 0.009* | | | |
| | (5.09) | (4.98) | (1.89) | | | |
| Media | | | | 0.111 | 0.021 * | 0.210 |
| | | | | (1.08) | (1.96) | (0.89) |
| Green × Post × Media | | | | 0.185 ** | 0.109 ** | 0.092 ** |
| | | | | (2.24) | (2.18) | (2.30) |
| Controls | YES | YES | YES | YES | YES | YES |
| Constant | −4.321 *** | −5.467 *** | −4.959 *** | −8.721 *** | −7.345 *** | −8.631 *** |
| | (−9.63) | (−8.69) | (−9.71) | (−7.68) | (−6.91) | (−7.02) |
| Observations | 24,858 | 24,858 | 24,858 | 24,858 | 24,858 | 24,858 |
| adj_$R^2$ | 0.612 | 0.598 | 0.584 | 0.592 | 0.601 | 0.588 |

**Notes**: *, **, *** Significant at 10, 5, and 1% levels, respectively; t values are reported in parentheses; refer to Table 1 for variables description.

### 4.4.2. Issuer with High Media Attention

In the information age, the media plays an important role. On the one hand, the media, as the intermediary of information dissemination, can disseminate the company's information in time, reducing market information asymmetry. At the same time, news reports are short and focused. This information paradigm is also conducive to information users' reading and absorbing of information and improves the efficiency of information dissemination. On the other hand, the media will integrate the confidence of different sources, such as relevant government policies, institutional analysis, public comments, etc., in reporting the reduction of the information search cost of information users. Therefore, when an enterprise issues green bonds, the media, as an intermediary of information dissemination and production, reports more related events, it can arouse more attention and interest of peer companies in learning, and it can more strongly promote the green technology innovation of peer companies. First, we sorted out the list of companies that successfully issued green bonds and then searched the Baidu search engine for "the name of the issuer, green bonds". Then, through keyword search, supplemented by manual reading, we found out the number of reports related to the issuance of green bonds by a specific company in the above-collected information. For example, we searched Baidu for "solar, green bonds" and then identified through reading that Sina Finance, Shanghai Securities News, Zhitong Finance Network, Liangjiang Finance Watch, and other media have reported on the green bonds issued by China Energy Conservation Solar Co., Ltd. (stock code: 00591) 12 times. We take the median of the total number of times the media report green bonds issued by enterprises in an industry as the demarcation point. Suppose the total number of times green bonds issued by enterprises in an industry are reported on a certain day is equal to or higher than the median. In that case, the industry is defined as an industry with high media attention. Other enterprises in the industry are assigned a value of 1; otherwise, it is assigned a value of 0. We put *Media* and *Green × Post × Media* into Model 1 simultaneously, and the results are shown in Table 10 columns 4–6. The coefficients of the interaction term *Green × Post × Media* are all significantly positive in the 5% confidence interval, which indicates that when the media attention of the issuer of green bonds is high, the green innovation of the peer company will be improved more significantly.

### 4.4.3. Close Competitors

In the product and capital markets, the company will pay close attention to its competitors and try to make the company superior to its competitors through imitation and other means. Brown et al. (2018) [63] found that when companies noticed that the stock exchange questioned their competitors' financial reports, they might improve their financial reports according to the inquiry letter so that they would not receive the inquiry letter. Therefore, we believe that the experience and income of green bond issuers will substantially impact the green technology innovation of close competitors. Referring to [63], we define the peer enterprises whose total assets do not exceed 10% of the green bond issuers as close competitors. If the peer company is a close competitor of the green bond issuer, *the Rival* is assigned to 1. Otherwise, it is 0. We put *Rival* and *Green × Post × Rival* into Model 1 simultaneously, and the results are shown in Table 11 columns 1–3. The coefficient of *Green × Post × Rival* is significantly positive at least at the level of 5%, indicating that when the peer enterprise and the green bond issuer are close competitors, they are more inclined to learn from the experience and benefits of issuers and improve their green innovation to try their best to make themselves competitive.

**Table 11.** Heterogeneity test results (b).

| Variable | (1) | (2) | (3) | (4) | (5) | (6) |
|---|---|---|---|---|---|---|
| | Close Competitors | | | In the Same Network of Directors | | |
| | *Patent1* | *Patent2* | *Patent3* | *Patent1* | *Patent2* | *Patent3* |
| Green × Post | 0.058 *** | 0.073 *** | 0.113 *** | 0.032 *** | 0.041 *** | 0.098 *** |
| | (3.98) | (3.62) | (3.27) | (4.56) | (4.02) | (3.99) |
| Rival | 0.008 | 0.000 | 0.010 | | | |
| | (0.21) | (1.10) | (1.23) | | | |
| Green × Post × Rival | 0.135 *** | 0.086 *** | 0.081 ** | | | |
| | (4.49) | (3.30) | (2.19) | | | |
| Director | | | | 0.011 * | 0.001 * | 0.023 ** |
| | | | | (1.92) | (2.00) | (2.23) |
| Green × Post × Director | | | | 0.203 ** | 0.186 *** | 0.116 * |
| | | | | (2.35) | (2.90) | (2.01) |
| Controls | YES | YES | YES | YES | YES | YES |
| Constant | −4.182 *** | −3.836 *** | −5.577 *** | −5.677 *** | −4.368 *** | −6.715 *** |
| | (−7.02) | (−7.16) | (−7.40) | (−9.12) | (−8.92) | (−9.72) |
| Observations | 24,858 | 24,858 | 24,858 | 24,858 | 24,858 | 24,858 |
| adj_R$^2$ | 0.673 | 0.652 | 0.791 | 0.612 | 0.633 | 0.698 |

**Notes**: *, **, *** Significant at 10, 5, and 1% levels, respectively; *t* values are reported in parentheses; refer to Table 1 for variables description.

### 4.4.4. In the Same Network of Directors

The director network, also known as director connection, director chain, board network, etc., refers to the direct and indirect connection established by individual directors of a company through serving on at least two boards of directors simultaneously. Previous studies have shown that the director network can bring imitation pressure, that is, the managers tend to imitate the behavior of other organizations in the same director network [65]. In addition, the director network is also a transmission channel of information and a learning platform of knowledge and market experience [66,67]; the experience of green bond issuers spreads faster and more thoroughly in the same network of directors. Therefore, we believe that when the peer company and the green bond issuers are in the same director network, the promotion effect of the peer company's green technology innovation is more significant. Drawing on the research of [68,69], this paper uses a virtual variable (Director) to represent the director chain enterprise. If the peer company and the green bond issuers are in the same director network, the value is 1. Otherwise, it is 0. We introduce the director chain as a moderating item into the Model 1, and the results are shown in Table 11 columns 4–6. The coefficient of interaction item (Green × Post ×

Director) is significant at least at a 10% level, which indicates that when the issuer of green bonds is in the same director network with the peer company, its spillover effect on the green innovation of the peer company is more vigorous.

## 5. Further Analysis

The foregoing analysis shows that peer enterprises in the same industry will improve their level of green innovation after learning about the experience and benefits of green bond issuers. We pay more attention to the impact of this result on the environmental performance of peer enterprises. To this end, we construct Model 3:

$$Environmental_{i,t+1} = \alpha_0 + \alpha_1 Green_i \times Post_t + \alpha_2 Patent_{i,t} + \alpha_3 Green_i \times Post_t \times Patent_{i,t} \\ + \sum \alpha_k Controls_{i,t} + \lambda_i + \delta_t + \varepsilon_{i,t} \tag{3}$$

Among them, *Environmental*$_{i,t+1}$ indicates the environmental performance of enterprise *i* in *t+1*. We use three indicators to measure environmental performance: (1) Environmental recognition (*E-recog*), referring to [70]; if the enterprise receives environmental recognition or other positive evaluation, it will be assigned a value of 1. Otherwise, it will be assigned a value of 0. (2) Environmental responsibility score (*E-score*), which we obtained from the evaluation of corporate social responsibility published by Hexun.com. We added 1 to the corporate environmental responsibility score in the actual calculation process and then took the natural logarithm. (3) Environmental certification (*E-certi*), which is consistent with [3]. If the enterprise's environmental management system has passed ISO14001 certification, the value is 1. Otherwise, it is 0. The environmental recognition and certification data come from the social responsibility database of the China Research Data Service Platform (CNRDS). From the empirical results in Table 12, after enterprises issue green bonds, peer enterprises in the same industry can get more environmental recognition or positive evaluation, their environmental responsibility scores are higher, and the probability of passing ISO14001 certification is also improved. The above evidence shows that green bonds' industrial technology spillover effect can produce positive environmental performance. This result is consistent with [2–4], that is, green bonds can indeed produce positive environmental benefits. This result also confirms our view that the reason why green bonds promote the overall environmental improvement in the early stage of development is that green bonds can promote the green innovation behavior of peer companies.

**Table 12.** Peer's environmental performance.

| Variable | (1) | (2) | (3) |
|---|---|---|---|
| | E-Recog | E-Score | E-Certi |
| Green × Post | 1.021 *** (2.99) | 0.179 *** (3.87) | 0.098 *** (4.02) |
| Controls | YES | YES | YES |
| Constant | 5.213 *** (8.77) | 8.245 *** (9.32) | 3.258 *** (6.98) |
| Observations | 24,858 | 24,858 | 24,858 |
| adj_R$^2$ | 0.579 | 0.601 | 0.632 |

**Notes**: *** Significant at 1% level; *t* values are reported in parentheses; refer to Table 1 for variables description.

## 6. Conclusions

Faced with global environmental problems, such as environmental pollution, resource depletion, and deterioration of the ecological environment, human beings have realized the harm of past production and consumption patterns to the environment. In order to better coordinate the relationship between economic development and ecological protection, sustainable development and green development have become the mainstream strategies of future economic development. Green development needs a lot of capital investment, but government funds can only cover a small part. Therefore, it is important to build a green financial system and guide social capital to participate in environmental governance. As a new financing tool with both "green" and "financial" characteristics, green bonds

are an effective way to help the green development strategy break through the capital shackles [6,10]. However, whether and how green bonds can play an active role in promoting the sustainable development of the environment has not been answered consistently. Against this background, we studied green bonds' industrial technology spillover effect, mechanism, and environmental performance. We find that after enterprises issue green bonds, peer companies will improve the quantity and quality of their green innovations based on the motivation of seeking benefits and avoiding harm, and this improvement effect is dynamic and sustainable. This conclusion is consistent with the research conclusion of [44], which states that corporate behavior motivated by organizational learning and reputation acquisition will be influenced by other corporate behaviors in the group. Different from the literature that suggests green bonds promote the green innovation of issuers, our research found another transmission path in which green bonds play a positive environmental role.

Then, we are curious about what kind of enterprises issuing green bonds can produce greater industry spillover effects. Industry leaders are the benchmark enterprises in the industry, and their behavior is often more considered and studied by other enterprises in the same industry. Therefore, when industry leaders issue green bonds, the information about green bonds issued by them is more likely to spread to peer enterprises, and it is also easier to cause peer companies to learn and imitate, resulting in greater industry spillover effect. At the same time, media reports are the main method of information diffusion and dissemination. Therefore, when enterprises issue green bonds with more media attention and publicity, it will naturally bring more industry spillover effects. In addition, enterprises tend to pay more attention to the behavior of close competitors. When competitors adopt new means or methods to obtain resources, enterprises will try to imitate or even surpass them. Therefore, the industry spillover effect will be more significant when the issuer of green bonds is a close competitor with its peer company. In addition, the board network is also an effective channel for information exchange. Therefore, when the issuer of green bonds is in the same board network as the peer company, the peer company's green innovation level is higher. These findings make us better understand the industry spillover effect of green bonds.

Finally, we are interested in whether peer enterprises' green innovation impacts their environmental performance. The answer to this question can echo the question we put forward in the introduction: at the early stage of the development of the green bond market, how can green bonds produce overall environmental performance when the number and coverage of green bonds issued are meager? We found that the green innovation behavior of peer enterprises promoted by green bonds can significantly improve their environmental performance, which is manifested in the higher probability of peer enterprises receiving environmental recognition or other positive evaluation, higher environmental responsibility score, and higher probability of passing ISO14001 certification. This makes us believe that the issuance of green bonds not only affects the pro-environmental behavior of the issuer [2] but also drives the peer enterprises to take more measures that are beneficial to environmental protection [3], resulting in overall environmental performance and social welfare.

In general, our research shows that the issuance of corporate green bonds can produce a good spillover effect of green innovation in the industry, and then produce positive environmental benefits, which is conducive to China's strategic goal of "carbon neutrality, carbon emission peak".

## 7. Theoretical Implications

Firstly, our research enriches the related literature on the factors influencing green innovation and helps to broaden the academic space of green innovation research. We enrich and improve the theoretical system of green innovation research. Secondly, this paper can make up for the deficiency of academic work in the research field of green finance theory and make positive academic contributions to the research of micro-transmission

mechanisms and the path of green finance. Thirdly, our research broadens the application scenario of peer learning theory and supplements the literature around financing decisions affecting the investment behavior of peer enterprises.

### 8. Practical Implications

Firstly, from the perspective of green bonds, we try to analyze the influence path of green finance on the green behavior of entities and provide evidence and experience for promoting ecologically sustainable development by market-oriented means. Secondly, we have confirmed the positive effect of green bonds on peer enterprises' green innovation and environmental performance. We affirmed the correct scheme to guide social capital to participate in environmental governance by developing the green bond market.

### 9. Suggestions

Green bond is a fixed-income tool that aims to provide financing for environmental and sustainable development projects, and it is attracting issuers and investors in various fields [1]. This article found the positive industrial spillover effect of green bonds, which showed that it was feasible to develop a green bond market, guide social capital to flow into green fields, and build a sustainable economy together. However, compared with the mission of green bonds, the scale of the green bond market is still tiny. Therefore, government departments should speed up the cultivation and construction of the green bond market and promote the orderly growth of green bond issuance. Future improvements can be made in the following aspects.

First, we agree with [1] and we recommend that government departments pursue the standardization of issuance by developing a standard green bond framework. At the same time, we recommend promoting the international convergence of this standard to attract more international capital to invest in Chinese green bonds.

Second, we suggest taking measures to encourage institutional investors, such as commercial banks, insurance companies, and securities companies, to invest in green bonds. For example, increasing the proportion of green bonds in financial institutions' green financial evaluation schemes, giving priority to selecting green bonds that meet the standards and include them in the central bank's pledge pool, making use of small tax spillovers to improve the return rate of investing in green bonds [71], and enriching and improving the diversified green bond product system to attract investors' attention and participation in the green bond market.

Third, we recommend enhancing green bond issuers' information transparency and promoting information circulation among enterprises. On the one hand, the relevant departments should establish a complete and transparent information disclosure framework for green bonds, standardize the information disclosure behavior of issuers, and promote the standardization and digitization of environmental benefit information disclosure of green bonds. On the other hand, China should further improve the third-party certification system, promote the healthy and orderly development of the external certification market, provide full access to the supervisory role of the third-party certification, and reasonably guarantee the quality of information disclosure.

Fourth, because of the media's attention to promoting the industrial spillover effect of the green bond, we suggest guiding the media to publicize the green bond and provide full access to the media's supervision and information diffusion. For example, we can choose typical green bond issuers to focus on tracking and reporting so that more enterprises and investors can know about green bonds. In addition, in view of the fact that the issuance of green bonds by industry leaders can produce greater industry spillover effects, qualified leading enterprises should be encouraged and guided to issue green bonds.

### 10. Limitations and Future Research

This study has three main limitations, which can be optimized in future research. Firstly, limited by data, this paper only considers the green patent for measuring an

enterprise's green technology innovation, while green innovation is a systematic process. In the future, we can participate in collecting relevant data and bringing green innovation input, efficiency, performance, and other factors into the research framework. Secondly, as a few samples of enterprises successfully issue green bonds at present, this paper does not subdivide the types of green bonds. In the future, we can further explore the influence of the heterogeneity of green bonds on the green technology innovation of peer enterprises. Thirdly, this paper does not consider the synergy mechanism of green bonds, that is, which factors can enhance the industry spillover effect of green bonds. In the future, we can supplement this vacancy from the perspectives of environmental regulation, good legal systems for intellectual property protection, and effective internal control environment. This paper also fails to consider the alternative mechanism of green bonds, that is, which factors can replace green bonds and produce the same positive effect as green bonds. In the future, we can further investigate whether other green financial products, such as green insurance, green funds, and green credit, can promote the green technological innovation of enterprises.

**Author Contributions:** Methodology, X.W.; Writing—original draft, J.L.; Writing—review & editing, D.B. and Y.B.; Funding acquisition, X.W. All authors have read and agreed to the published version of the manuscript.

**Funding:** This research was funded by [Chongqing Federation Social Science Planning Project] grant number [2021NDYB079], [SiChuan International Studies University School-level Scientific Research Project] grant number [sisu202134], [Youth Project of Science and Technology Research Program of Chongqing Education Commission] grant number [KJQN202200902], [SiChuan International Studies University School-level Scientific Research Project] grant number [sisu202043], And The APC was funded by [2021NDYB079].

**Institutional Review Board Statement:** Not applicable.

**Informed Consent Statement:** Not applicable.

**Data Availability Statement:** The data involved in this study mainly come from the following three databases: CNRDS https://www.cnrds.com/Home/Login, accessed on 10 December 2022 CSMAR https://www.gtarsc.com/, accessed on 10 December 2022. Wind https://www.wind.com.cn/, accessed on 10 December 2022.

**Conflicts of Interest:** The authors declare no conflict of interest.

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
