# Peer review of "Green Bond Issuance and Peer Firms’ Green Innovation"

_sustainability, doi:10.3390/su142417035_

Round 1

Reviewer 1 Report

- row 22. suggest you use a commonly accepted definition of green bonds at the start of your paper with a well accepted source. this will be easier for the reader

- row 40-45. what you mean by "economic consequences"is not well explained in the text

- row 59. "wind" database? what is "wind", you refer to it a few times in the paper

- suggest to incorporate findings from the following paper to discuss potential market growth of green bonds, in the existing literature alongside Flammer: What Future for the Green Bond Market? How Can Policymakers, Companies, and Investors Unlock the Potential of the Green Bond Market? Pauline Deschryver and Frederic de Mariz

- generally speaking, you need to better explain the data sources and you build the variables, and i would do that earlier in the paper

- row 61. "Our research scope is 61 limited to the green bond market in China, which is conducive to eliminating the influence of national institutional 62 factors on the research conclusion." On the contrary, a cross section would have allowed to eliminate country variables. i would remove that sentence

- "green innovation", "green technology", you mix up the concepts throughout the paper. choose one concept, define it and stick to it.row 232, what is the source for this definition of green tech innovation? row 244-245, those 3 types of patents are listed in the IPC codes? 244-245, not clear why you Patent1, Patent2, Patent2, while those are three categories. the definitions / differences btw those 3 categories is also not clear

- row 65. at a minimum, you need to explain what is the PSM-DID model, what does it do? why it is appropriate to solve this problem?

- row 78. what is "board network"?

- row 112. i would not put the key research question only on row 112, suggest to move this earlier in the paper

- row 208. how do you define the concept of "same industry"

- row 215-216. you need to explain more on CNRDS and CSMAR, the data is key to your research conclusion. you also need to explain why / how often you had to go back to companies' annual reports

- it would be very interesting to show what industries have more or less spillover effects. this would be quite valuable to policymakers

- you have a number of other tests that you apply. what are those trying to achieve? should this go into an appendix? they are valuable, but they do break the flow of your argument

- row 534 "approval procedures of green bond issuance". i would disagree, there is a very significant risk of greenwashing and relaxing rules / supervision is not a policy route. if anything raising the bar will probably attract more investor demand and lead to more issuances 

- row 549. no need to build a unified verification system for the country, most market participants use ICMA-aligned rules

- row 563. not clear what you mean by synergy and substitution

Reviewer 2 Report

This study examines the impact of green bond issuance on peer firms’ green innovation. The results show that the issuance of corporate green bonds can significantly promote the quantity and quality of peer firms’ green 10 innovation, and this promotion effect is sustainable.

Comments:

·         I would advise authors to stress the originality and the value-added of the paper. Although the study is timely, I think the Introduction section needs to be reinforced by addressing the article’s uniqueness and novelty.

·         The theoretical background is appropriately developed and discussed. Specifically, Section 2 provides an excellent overview of the framework for analysis. The authors could find other useful and interesting aspects in some recent work, so please explore:

-          Tsagkanos, A., Sharma, A., & Ghosh, B. (2022). Green Bonds and Commodities: A New Asymmetric Sustainable Relationship. Sustainability, 14(11), 6852.

-          Cicchiello, A. F., Cotugno, M., Monferrà, S., & Perdichizzi, S. (2022). Which are the factors influencing green bonds issuance? Evidence from the European bonds market. Finance Research Letters, 50, 103190.

-          Deng, J., Lu, J., Zheng, Y., Xing, X., Liu, C., & Qin, T. (2022). The Impact of the COVID-19 Pandemic on the Connectedness between Green Industries and Financial Markets in China: Evidence from Time-Frequency Domain with Portfolio Implications. Sustainability, 14(20), 13178.

-          Cicchiello, A. F., Cotugno, M., Monferrà, S., & Perdichizzi, S. (2022). Credit spreads in the European green bond market: A daily analysis of the COVID‐19 pandemic impact. Journal of International Financial Management & Accounting.

-          Taghizadeh-Hesary, F., Yoshino, N., & Phoumin, H. (2021). Analyzing the characteristics of green bond markets to facilitate green finance in the post-COVID-19 world. Sustainability, 13(10), 5719.    

·         Implications at practical and theoretical levels should be further strengthened following the paper’s discussion.

·         The conclusion section looks weak. Just as the introduction gives the first impression to the reader, the conclusion offers a chance to leave a lasting impression. I suggest the authors strengthen this section by highlighting the importance of their idea, summarizing their thoughts and conveying the more significant implications of the study. Limitations of the study should also be included.

·         In general, the manuscript is well written. Some minor spelling errors to be corrected
